# Improving Adaptive Moment Optimization via Preconditioner Diagonalization

## Abstract

Modern deep learning heavily relies on adaptive optimization methods like Adam and its variants, celebrated for their robustness against model scale and ease of hyperparameter tuning. However, the gradient statistics employed by these methods often do not leverage sufficient gradient covariance information, leading to suboptimal updates in certain directions of the parameter space and potentially slower convergence. In this work, we keep track of such covariance statistics in the form of a structured preconditioner matrix. Unlike other works, our approach does not apply direct approximations to estimate this matrix. We instead *implement an invertible transformation that maps the preconditioner matrix into a new space where it becomes approximately diagonal*. This enables a diagonal approximation of the preconditioner matrix in the transformed space, offering several computational advantages. Empirical results show that our approach can substantially enhance the convergence speed of modern adaptive optimizers. Notably, for large language models like LLaMA, we can achieve a **2x** speedup in sample efficiency compared to Adam. In addition, our method can also be integrated with memory-efficient optimizers like Adafactor to manage computational overhead.

## 1. Introduction

In the realm of deep learning optimization, finding efficient and reliable solutions to complex problems has become a central challenge. As model scales and datasets continue to expand, such optimization problems usually demand extensive training time and substantial computational resources to achieve state-of-the-art performances.

Standard first-order methods, such as stochastic gradient descent (SGD) and its variants, have emerged as canonical tools for training large-scale deep networks. These methods are straightforward to implement using modern automatic differentiation frameworks and are easily adaptable to non-conventional training setups (Konečnỳ et al., 2016; Li et al., 2020; Finn et al., 2017). However, despite their strong theoretical grounding (Şimşekli et al., 2019; Zhou et al., 2020; Smith et al., 2021; Tian et al., 2023), first-order methods typically require meticulous tuning of hyperparameters to ensure the optimization process can converge to the desired local optima. In practice, these methods often struggle when navigating highly non-convex loss surfaces, a common characteristic of deep learning models. Pathological features like saddle points, flat regions, and sharp valleys in the loss landscape can significantly hinder convergence, leading to inefficient training.

To tackle these challenges, optimization techniques have evolved to incorporate curvature geometry or second-order information, providing more adaptive and efficient updates. A classic family of such algorithms is preconditioned gradient methods, in which the gradient is premultiplied by a matrix called a preconditioner before each optimization step. Classic algorithms in this family include Newton methods (Bonnans et al., 2006) and Natural Gradient (Martens, 2020), which employ the inverses of local Hessian and Fisher Information Matrix as preconditioners, respectively. Although preconditioning methods typically exhibit much faster convergence than first-order approaches, their practical application is limited by the size of most real-world problems, as they demand quadratic storage and cubic computation time for each gradient update (Fletcher, 2000; Bonnans et al., 2006).

In addition to preconditioning, adaptive moment estimation is another line of work that has been highly influential in deep learning optimization. These methods, including AdaGrad (Duchi et al., 2011), Adam (Kingma & Ba, 2014), AMSGrad (Reddi et al., 2019), and Adafactor (Shazeer & Stern,

2018) dynamically adapt the learning rate of each parameter throughout the optimization process by leveraging cumulative second-order gradient statistics. While their theoretical foundations are not yet fully explored, these algorithms have demonstrated more robust empirical results than SGD in various domains and often exhibit better convergence behaviors in practice (Loshchilov & Hutter, 2017; Zhang et al., 2020).

**Our approach**. In this work, we explore adaptive moment-based algorithms from the perspective of preconditioning methods. We are driven by the understanding that the second-moment estimates in these algorithms can be derived from the diagonal approximation of a structured preconditioner. We propose to improve this approximation by applying an invertible transformation that maps the preconditioner into a new space where it becomes approximately diagonal. In this transformed space, the diagonal elements of the preconditioner can be accumulated to estimate second-order statistics better. Since the transformation is invertible, we can formulate the update on the original parameter space by doing a simple projection back.

Our key contributions are outlined as follows:

1. Our approach is designed to be both straightforward and versatile, facilitating easy integration into existing adaptive moment-based optimizers such as Adam, AMSGrad, Adafactor, and variants.

2. We establish a convergence guarantee for the general framework without requiring typical strong assumptions. This guarantee is significant because it is broadly applicable to a wide range of adaptive optimizers, ensuring reliable performances in diverse scenarios.

3. Empirical results show that our proposed methods can substantially enhance the convergence speed and efficiency of adaptive moment-based optimization baselines. Particularly in pretraining large-scale models like LLaMA, our approach can achieve a speedup of 1.5x to 2x compared to the Adam, but with manageable computational overhead.

**Notations**: For any matrices $\mathbf{A}, \mathbf{B}$ of size $m \times n$, we use $\sqrt{\mathbf{A}}$ for element-wise square root, $\mathbf{A}^2$ for element-wise square, and $\mathbf{A}/\mathbf{B}$ for element-wise division. The symbol $\mathbf{A}^\top$ stands for the transpose matrix, $\langle \mathbf{A}, \mathbf{B} \rangle = \text{trace}(\mathbf{A}^\top \mathbf{B})$ represents the inner products of matrices, and $\mathbf{A} \otimes \mathbf{B}$ is the Kronecker product. Let $\text{diag}(.)$ denote the diagonal matrix, and $\text{vec}(.)$ denote the vectorization of a matrix. We write $[t] = \{1, \ldots, t\}$ as the first $t$ positive integers.

## 2. PRELIMINARIES AND BACKGROUND

We consider an unconstrained, continuous optimization problem $\min_{\mathbf{W} \in \mathbb{R}^d} \mathcal{L}(\mathbf{W}; \mathbf{X})$, with $\mathbf{X}$ denotes observations, $\mathcal{L} : \mathbb{R}^d \to \mathbb{R}$ is a proper differentiable and lower bounded objective function.

### 2.1. PRECONDITIONED GRADIENT DESCENT

The iterative scheme of preconditioned gradient descent can be expressed as follows:

$$\mathbf{W}_{t+1} = \mathbf{W}_t - \eta_t \mathbf{C}(t) \nabla \mathcal{L}(\mathbf{W}_t; \mathbf{X}), \tag{1}$$

where the matrix $\mathbf{C}(t)$ is referred to as preconditioner. When $\mathbf{C}(t)$ is set to the identity matrix, the update above simplifies to ordinary gradient descent. To capture curvature informativeness, systematic designs of $\mathbf{C}(t)$ have been developed using local numerical approximations. Classic algorithms in this category, including Newton methods and Natural Gradient, utilize the inverse of Hessian and Fisher Information Matrix, respectively, as preconditioners. These methods offer a built-in mechanism for curvature awareness, promoting larger updates in directions associated with small Hessian eigenvalues to swiftly navigate flat regions while limiting movement in directions with large Hessian eigenvalues to avoid sharp valleys. However, for large-scale models, further approximations to the preconditioners are necessary to ensure their practicality. Various techniques have been proposed for this purpose, such as Quasi-Newton methods (Fletcher, 2000), Gaussian-Newton estimators (Botev et al., 2017; Martens, 2020; Liu et al., 2023), K-FAC (Martens & Grosse, 2015; Grosse & Martens, 2016).

## 2.2. ADAPTIVE MOMENT ESTIMATION

These methods, such as AdaGrad, Adam, AMSGrad dynamically adjust learning rates for each parameter by incorporating a form of gradient-based statistics. Specifically with Adam, it estimates both first and second-order moments by maintaining exponential moving averages (EMA) on the mean and uncentered variance of gradient information across iterations. The update rules are:

$$\mathbf{M}_t = \hat{\beta}_{1t}\mathbf{M}_{t-1} + (1 - \hat{\beta}_{1t})\nabla\mathcal{L}(\mathbf{W}_t; \mathbf{X}) \triangleq \underset{\tau \in [t]}{\mathrm{EMA}}\left[\nabla\mathcal{L}(\mathbf{W}_\tau; \mathbf{X})\right]$$

$$\mathbf{V}_t = \hat{\beta}_{2t}\mathbf{V}_{t-1} + (1 - \hat{\beta}_{2t})\nabla\mathcal{L}(\mathbf{W}_t; \mathbf{X})^2 \triangleq \underset{\tau \in [t]}{\mathrm{EMA}}\left[\nabla\mathcal{L}(\mathbf{W}_\tau; \mathbf{X})^2\right]$$

$$\mathbf{W}_{t+1} = \mathbf{W}_t - \eta_t \frac{\mathbf{M}_t}{\sqrt{\mathbf{V}_t} + \epsilon}$$

where $\hat{\beta}_{1t}, \hat{\beta}_{2t}$ are the decay moment coefficients, $\epsilon$ is the smoothing tolerance constant to avoid numerical instability. In principle, the first moment amplifies the gradient in directions that are consistently the same sign and dampens the gradient in directions that are reversing sign. Meanwhile, the second moment captures the curvature by adjusting the step size based on gradient magnitude: smaller steps in steep-gradient regions to avoid overshooting and larger steps in shallow-gradient regions for faster convergence.

## 2.3. ADAPTIVE MOMENT ESTIMATION VIA DIAGONAL PRECONDITIONING APPROXIMATION

We examine the matrix case where $\mathbf{W}$ represents a weight parameter with dimensions $m \times n$. Let $\mathbf{G}_t$ denote the gradient of loss function $\nabla\mathcal{L}(\mathbf{W}_t; \mathbf{X})$ at iteration $t$. We analyze a preconditioner $\mathbf{C}_t$ exploiting the second-order moment of accumulated gradients in the following *inverse* form:

$$\mathbf{C}_t = \left[\mathbb{E}_{p(\mathbf{X})}[\mathrm{vec}(\mathbf{G}_t)\mathrm{vec}(\mathbf{G}_t)^\top] + \epsilon\mathbf{I}_{mn}\right]^{-1/2} \tag{2}$$

We have $\mathbf{C}_t$ as a positive definite matrix of size $mn \times mn$, which is quadratic to the size of model parameter $\mathbf{W}$. An analytical formulation of this quality is often intractable in practice. However, under the assumption of stationary gradient distribution, we can approximate the expectation by leveraging minibatch sampling in conjunction with the exponential moving average technique. We then obtain an empirical preconditioner defined by:

$$\mathbf{C}_t = \left[\underset{\tau \in [t]}{\mathrm{EMA}}\left[\mathrm{vec}(\mathbf{G}_\tau)\mathrm{vec}(\mathbf{G}_\tau)^\top\right] + \epsilon\mathbf{I}_{mn}\right]^{-1/2}, \tag{3}$$

where $\mathbf{G}_\tau := \frac{1}{|\mathcal{B}_\tau|}\sum_{\mathbf{X} \in \mathcal{B}_\tau}\nabla\mathcal{L}(\mathbf{W}_\tau; \mathbf{X})$ is the minibatch gradient at training step $\tau$. This empirical preconditioner closely resembles the full matrix version of AdaGrad (Duchi et al., 2011), but instead of using a cumulative sum, we apply an exponential moving average (EMA).

Directly calculating and storing the matrix $\mathbf{C}_t$ is computationally expensive, particularly with modern network architectures, since it requires inverting a very large matrix. A practical way to alleviate this bottleneck is by using diagonal approximation:

$$\mathbf{C}_t^{(d)} = \left[\underset{\tau \in [t]}{\mathrm{EMA}}\,\mathrm{diag}\left(\mathrm{vec}(\mathbf{G}_\tau)\mathrm{vec}(\mathbf{G}_\tau)^\top\right) + \epsilon\mathbf{I}_{mn}\right]^{-1/2}$$

$$= \left[\underset{\tau \in [t]}{\mathrm{EMA}}\,\mathrm{diag}\left(\mathrm{vec}(\mathbf{G}_\tau^2) + \epsilon\right)\right]^{-1/2} \tag{4}$$

$$= \mathrm{diag}^{-1/2}\left(\underset{\tau \in [t]}{\mathrm{EMA}}\left[\mathrm{vec}(\mathbf{G}_\tau^2) + \epsilon\right]\right).$$

The diagonal preconditioner $\mathbf{C}_t^{(d)}$ represents the inverse root square of the second-order gradient accumulator, which is widely adopted as the adaptive moment estimation in optimizers such as Ada-Grad, RMSprop, Adam, and variants. Implementing this diagonal approximation offers advantages in both computational efficiency and memory usage. Amari et al. (2019) also demonstrate that the off-diagonal components of $\mathbf{C}_t$ are smaller than the diagonal components by a factor of $1/\sqrt{N}$, where $N$ is the number of elements in the matrix. This insight contributes to understanding the practical success of optimizers like AdaGrad, Adam, and others. However, by omitting the off-diagonal elements, the algorithm does not incorporate gradient correlations, which can be particularly useful in accelerating optimization (Martens & Grosse, 2015; Gupta et al., 2018; Liu et al., 2023).

---

**Algorithm 1** AdaDiag and AdaDiag++ for matrix parameter $\mathbf{W}$ of size $m \times n$, $m \leq n$

---

**Inputs:** moment coefficients $\beta_1, \beta_2$, smoothing term $\epsilon = 10^{-8}$, regularization constant $\lambda$
**Initialization:** weight parameters $\mathbf{W}_0 \in \mathbb{R}^{m \times n}$, initial moments $\mathbf{M}_0, \mathbf{V}_0 \leftarrow 0$
**repeat**
    $t \leftarrow t + 1$
    $\mathbf{G}_t = \nabla \mathcal{L}(\mathbf{W}_t; \mathcal{B}_t)$
    **if** $t \bmod T = 0$ **then**
        $\mathbf{P}_t, \_, \mathbf{Q}_t^\top = \texttt{torch.linalg.svd}(\mathbf{G}_t, \texttt{full\_matrices=True})$
    **else**
        $\mathbf{P}_t, \mathbf{Q}_t^\top \leftarrow \mathbf{P}_{t-1}, \mathbf{Q}_{t-1}^\top$
    **end if**
    $\widetilde{\mathbf{G}}_t = \mathbf{P}_t^\top \mathbf{G}_t \qquad \widetilde{\mathbf{G}}_t = \mathbf{P}_t^\top \mathbf{G}_t \mathbf{Q}_t$
    $\mathbf{M}_t = \hat{\beta}_{1t} \mathbf{M}_{t-1} + (1 - \hat{\beta}_{1t}) \widetilde{\mathbf{G}}_t$
    $\mathbf{V}_t = \hat{\beta}_{2t} \mathbf{V}_{t-1} + (1 - \hat{\beta}_{2t}) \widetilde{\mathbf{G}}_t^2$
    $\mathbf{W}_{t+1} = \mathbf{W}_t - \eta_t \left( \mathbf{P}_t \dfrac{\mathbf{M}_t}{\sqrt{\mathbf{V}_t} + \epsilon} + \lambda \mathbf{W}_t \right) \qquad \mathbf{W}_{t+1} = \mathbf{W}_t - \eta_t \left( \mathbf{P}_t \dfrac{\mathbf{M}_t}{\sqrt{\mathbf{V}_t} + \epsilon} \mathbf{Q}_t^\top + \lambda \mathbf{W}_t \right)$
**until** *stopping criterion is met*
**return** optimized parameter $\mathbf{W}_t$

---

## 3. PRECONDITIONER DIAGONALIZATION WITH GRADIENT PROJECTION

To leverage the off-diagonal components of the preconditioner matrix, one can implement structural approximations, like Gaussian-Newton estimators (Botev et al., 2017; Martens, 2020; Liu et al., 2023), or Kronecker factorization (Martens & Grosse, 2015; Gupta et al., 2018). In this section, we approach the problem from a different perspective of preconditioner diagonalization. Specifically, we will rationalize the diagonal approximation assumption by applying an implicit orthogonal transformation on the preconditioner matrix $\mathbf{C}_t$. Intuitively, this technique will rotate the gradients to align with coordinate axes partially, ultimately causing the matrix $\mathbf{C}_t$ to become approximately diagonal. Moreover, we will show that this transformation is invertible via a network reparameterization, leading to a simple update on the original parameter space.

Recall $\mathbf{G}_\tau$ is a matrix of size $m \times n$, with $m \leq n$. Define $\mathcal{C}(\mathbf{G}_\tau) \triangleq \text{vec}(\mathbf{G}_\tau)\text{vec}(\mathbf{G}_\tau)^\top$, then:

$$\mathbf{C}_t = \left[ \underset{\tau \in [t]}{\text{EMA}} \left[ \mathcal{C}(\mathbf{G}_\tau) \right] + \epsilon \mathbf{I}_{mn} \right]^{-1/2}. \tag{5}$$

Let's start by drawing some intuitions through the diagonalization of matrix $\mathcal{C}(\mathbf{G}_\tau)$. Given the special formula of $\mathcal{C}(\mathbf{G}_\tau)$, we can perform a straightforward approach using Singular Value Decomposition (SVD) on the gradient $\mathbf{G}_\tau$. Suppose we have:

$$\mathbf{G}_\tau = \mathbf{P}_\tau \mathbf{\Sigma}_\tau \mathbf{Q}_\tau^\top$$

in which $\mathbf{P}_\tau, \mathbf{Q}_\tau$ are orthogonal matrices of size $m \times m, n \times n$, respectively, and $\mathbf{\Sigma}_\tau$ is a diagonal matrix of size $m \times n$. Substituting this representation into $\mathcal{C}(\mathbf{G}_\tau)$ gives us:

$$\mathcal{C}(\mathbf{G}_\tau) = \text{vec}(\mathbf{P}_\tau \mathbf{\Sigma}_\tau \mathbf{Q}_\tau^\top)\text{vec}(\mathbf{P}_\tau \mathbf{\Sigma}_\tau \mathbf{Q}_\tau^\top)^\top = (\mathbf{Q}_\tau \otimes \mathbf{P}_\tau)\text{vec}(\mathbf{\Sigma}_\tau)\text{vec}(\mathbf{\Sigma}_\tau)^\top (\mathbf{Q}_\tau \otimes \mathbf{P}_\tau)^\top \tag{6}$$

Since $\mathbf{\Sigma}_\tau$ is a diagonal matrix, we have $\text{vec}(\mathbf{\Sigma}_\tau)\text{vec}(\mathbf{\Sigma}_\tau)^\top$ is almost diagonal (off-diagonal elements are mostly zero). Moreover, the matrix $\mathbf{Q}_\tau \otimes \mathbf{P}_\tau$ satisfies $(\mathbf{Q}_\tau \otimes \mathbf{P}_\tau)(\mathbf{Q}_\tau \otimes \mathbf{P}_\tau)^\top = (\mathbf{Q}_\tau \mathbf{Q}_\tau^\top) \otimes (\mathbf{P}_\tau \mathbf{P}_\tau^\top) = \mathbf{I}_{mn}$, therefore we can consider $\mathbf{Q}_\tau \otimes \mathbf{P}_\tau$ as an orthogonal diagonalizing matrix with:

$$(\mathbf{Q}_\tau \otimes \mathbf{P}_\tau)^{-1} \mathcal{C}(\mathbf{G}_\tau)(\mathbf{Q}_\tau \otimes \mathbf{P}_\tau) = \text{vec}(\mathbf{\Sigma}_\tau)\text{vec}(\mathbf{\Sigma}_\tau)^\top.$$

Alternatively, the diagonalization process above can be equivalently derived from $\mathcal{C}(\widetilde{\mathbf{G}}_\tau)$, with $\widetilde{\mathbf{G}}_\tau \triangleq \mathbf{P}_\tau^\top \mathbf{G}_\tau \mathbf{Q}_\tau = \mathbf{\Sigma}_\tau$. This rotation aligns the gradient $\widetilde{\mathbf{G}}_t$ with coordinate axes and consequently induces a roughly diagonal structure on $\mathcal{C}(\widetilde{\mathbf{G}}_\tau)$.

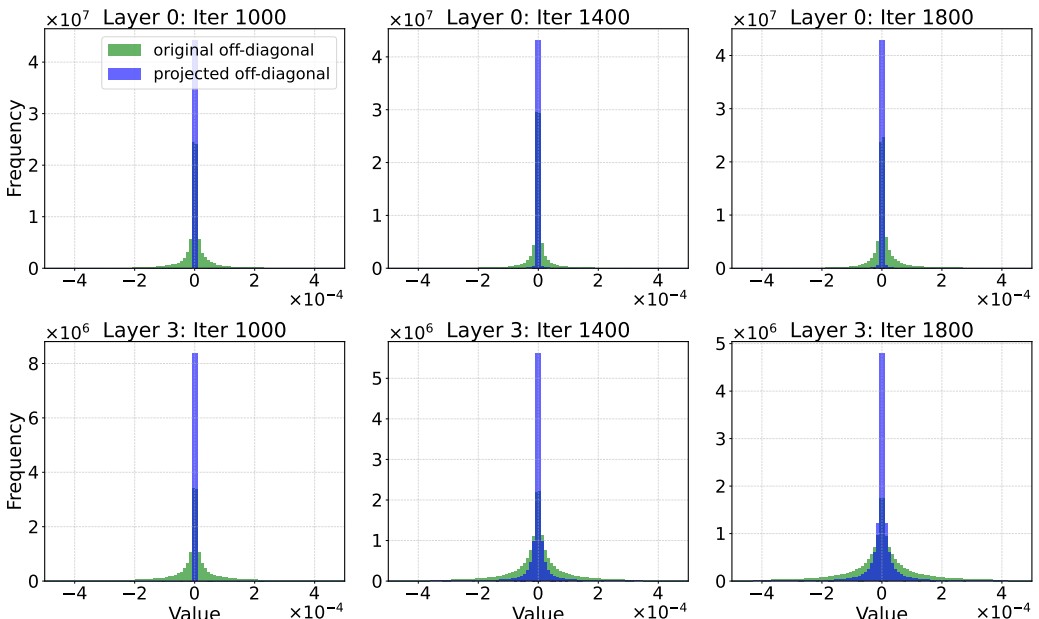

Figure 1: Histograms of off-diagonal elements $\mathcal{C}(\mathbf{G}_\tau)$ (original) and $\mathcal{C}(\widetilde{\mathbf{G}}_\tau)$ (*two-sided* projection), corresponding to the two first layers of ResNet50 trained on ImageNet1k. In this experiment, we set the frequency $T = 500$ and plot histograms at iterations with and without SVD applied.

## 3.1. PERIODIC PROJECTION ONTO GRADIENT SUBSPACES

The analysis presented above applies only to the iteration in which the SVD is implemented. However, it is impractical to utilize SVD at every iteration of the training procedure due to computational overhead. Fortunately, there are recent works on low-rank gradient projection (Gur-Ari et al., 2018; Gooneratne et al., 2020; Zhao et al., 2024; Liang et al., 2024) indicating that the optimization process usually acts on low-dimensional subspaces. GaLore (Zhao et al., 2024) exploits this concept by showing that the training trajectory can be divided into continual subspaces, from which the gradients within each subspace can be governed by a common spanning basis. GaLore deployed this idea by periodically applying SVD on the gradients to extract projection matrices. Mathematically, during each period of length $T$, say $[\kappa T, (\kappa + 1)T]$, the gradient $\mathbf{G}_\tau$ can be represented via SVD as:

$$\mathbf{G}_\tau \approx \mathbf{P}_\kappa \mathbf{\Sigma}_\tau \mathbf{Q}_\kappa^\top \quad \forall \tau \in [\kappa T, (\kappa + 1)T]$$

where $\mathbf{P}_\kappa, \_, \mathbf{Q}_\kappa^\top \leftarrow \text{SVD}(\mathbf{G}_{\kappa T})$ are kept the same throughout the period.

We can adapt this assumption to our framework. In this way, we can expect that our projected gradients in each period are still approximately diagonal, namely:

$$\widetilde{\mathbf{G}}_\tau \triangleq \mathbf{P}_\kappa^\top \mathbf{G}_\tau \mathbf{Q}_\kappa \approx (\mathbf{P}_\kappa^\top \mathbf{P}_\kappa) \mathbf{\Sigma}_\tau (\mathbf{Q}_\kappa^\top \mathbf{Q}_\kappa) = \mathbf{\Sigma}_\tau.$$

As a result, we can achieve a desired diagonal structure on $\mathcal{C}(\widetilde{\mathbf{G}}_\tau), \forall \tau \in [\kappa T, (\kappa + 1)T]$.

**Why the full matrices $\mathbf{P}_\kappa, \mathbf{Q}_\kappa$ matter: connection and difference with Galore**. It is essential to note that, since GaLore's focus is on memory efficiency, they just implemented truncated SVD to capture the top-$K$ representation of the gradient matrices. This assumption is reasonable when considering that gradients are low-rank matrices. However, using truncated SVD involves careful tweaking of the $K$ values and more importantly, the gradient projection step has to be executed on the smaller dimension of the matrix.[1] On the other hand, our framework would require sophisticated

---

[1]Let's omit subscripts $\tau, \kappa$ for simplicity. Consider $\mathbf{G}_{m\times n} \approx \mathbf{P}_{m\times K} \mathbf{\Sigma}_{K\times K} \mathbf{Q}_{n\times K}^\top$ as the truncated SVD of gradient matrix $\mathbf{G}_{m\times n}$ with $m \leq n$. The GaLore algorithm performs a gradient projection as $\widetilde{\mathbf{G}} = \mathbf{P}_{m\times K}^\top \mathbf{G}$, which maps $n$ vectors of size $m$ from $\mathbf{G}$ onto the subspace spanned by $K$ vectors of size $m$ from $\mathbf{P}_{m\times K}$. Since $K \leq m \leq n$, this operator is more effective than projecting in larger dimensions, i.e. $\widetilde{\mathbf{G}} = \mathbf{G}\mathbf{Q}_{n\times K}$.

modifications in GaLore, of which $\mathbf{P}_\kappa, \mathbf{Q}_\kappa$ need to be the full matrices or $K$ is at least the effective rank. This guarantees that $\mathbf{P}_\kappa, \mathbf{Q}_\kappa$ can form complete bases for the rows and columns of the gradient matrices in each period. To make the algorithm effortless, we propose to adopt the full matrices instead of tunning effective rank $K$.

**One-sided projection**. Instead of using *two-sided* projection as described so far, we can opt for a simpler version involving just *one-sided* projection, namely $\widetilde{\mathbf{G}}_\tau \triangleq \mathbf{P}_\tau^\top \mathbf{G}_\tau = \mathbf{\Sigma}_\tau \mathbf{Q}_\tau^\top$, then we have:

$$\mathcal{C}(\widetilde{\mathbf{G}}_\tau) \triangleq \mathrm{vec}(\widetilde{\mathbf{G}}_\tau)\mathrm{vec}(\widetilde{\mathbf{G}}_\tau)^\top = \mathrm{vec}(\mathbf{\Sigma}_\tau \mathbf{Q}_\tau^\top)\mathrm{vec}(\mathbf{\Sigma}_\tau \mathbf{Q}_\tau^\top)^\top$$
$$= (\mathbf{Q}_\tau \otimes \mathbf{I}_m)\mathrm{vec}(\mathbf{\Sigma}_\tau)\mathrm{vec}(\mathbf{\Sigma}_\tau)^\top (\mathbf{Q}_\tau \otimes \mathbf{I}_m)^\top. \qquad (7)$$

The projected gradient $\widetilde{\mathbf{G}}_\tau = \mathbf{\Sigma}_\tau \mathbf{Q}_\tau^\top$ inherently exhibits a sparse structure as illustrated in Figure 2. This is because $\mathbf{\Sigma}_\tau$ is a diagonal matrix and the smallest singular values on the diagonal will zero out the magnitude of corresponding rows on $\mathbf{Q}_\tau$. In Figure 1 and 9, we further show the histograms of off-diagonal elements of $\mathcal{C}(\mathbf{G}_\tau)$ and $\mathcal{C}(\widetilde{\mathbf{G}}_\tau)$ (both *two-sided* and *one-sided*), corresponding to the two first layers of ResNet50 trained on ImageNet1k. We can observe notable differences in sparsity patterns of off-diagonal elements of $\mathcal{C}(\widetilde{\mathbf{G}}_\tau)$ compared to the original $\mathcal{C}(\mathbf{G}_\tau)$ over iterations. The matrix $\mathcal{C}(\widetilde{\mathbf{G}}_\tau)$ is roughly diagonal, enabling a more accurate diagonal approximation for the preconditioner matrix $\widetilde{\mathbf{C}}_t$.

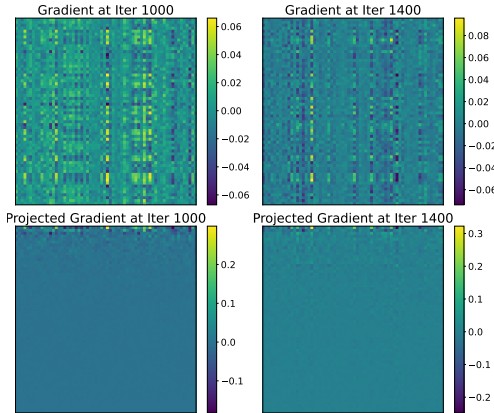

Figure 2: Sparsity of *one-sided* projection.

### 3.2. GRADIENT PROJECTION IMPLIES NETWORK REPARAMETERIZATION

In the previous sections, we demonstrated that we can rotate the gradients $\mathbf{G}_\tau$ to $\widetilde{\mathbf{G}}_\tau$ in such a way that the matrix $\mathcal{C}(\widetilde{\mathbf{G}}_\tau)$ is approximately diagonal. Consequently, it induces a diagonal approximation of $\widetilde{\mathbf{C}}_t$ as follows:

$$\widetilde{\mathbf{C}}_t^{(d)} = \left[\underset{\tau \in [t]}{\mathrm{EMA}} \, \mathrm{diag}\left[\mathrm{vec}(\widetilde{\mathbf{G}}_\tau)\mathrm{vec}(\widetilde{\mathbf{G}}_\tau)^\top\right] + \epsilon\mathbf{I}_{mn}\right]^{-1/2}$$
$$= \left[\underset{\tau \in [t]}{\mathrm{EMA}} \, \mathrm{diag}\left[\mathrm{vec}(\widetilde{\mathbf{G}}_\tau^2) + \epsilon\right]\right]^{-1/2} \qquad (8)$$
$$= \mathrm{diag}^{-1/2}\left[\underset{\tau \in [t]}{\mathrm{EMA}}\left[\mathrm{vec}(\widetilde{\mathbf{G}}_\tau^2) + \epsilon\right]\right],$$

This makes the preconditioned gradient at iteration $t$ becomes:

$$\widetilde{\mathbf{C}}_t^{(d)}\mathrm{vec}(\widetilde{\mathbf{G}}_t) = \frac{\mathrm{vec}(\widetilde{\mathbf{G}}_t)}{\sqrt{\mathrm{EMA}_{\tau \in [t]}\left[\mathrm{vec}(\widetilde{\mathbf{G}}_\tau^2) + \epsilon\right]}} \qquad (9)$$

However, this quantity cannot be directly applied to update the weight parameters, as the gradient projection implicitly imposes a reparameterization on the weight space. In Figure 3, we illustrate how the update in equation 9 can be utilized to learn a rotated network rather than the original one. This rotated network introduces two auxiliary layers defined by $\mathbf{P}_\kappa, \mathbf{Q}_\kappa^\top$, and reparameterizes the original weight parameters as $\widetilde{\mathbf{W}} = \mathbf{P}_\kappa^\top \mathbf{W}\mathbf{Q}_\kappa$. While this transformation does not alter the forward pass of the original network, it does lead to a corresponding gradient, represented as:

$$\nabla_{\widetilde{\mathbf{W}}}\mathcal{L}(\widetilde{\mathbf{W}};\mathbf{X}) = \mathbf{P}_\kappa^\top \nabla_{\mathbf{W}}\mathcal{L}(\widetilde{\mathbf{W}};\mathbf{X})\mathbf{Q}_\kappa \underset{\text{same forward response}}{=} \mathbf{P}_\kappa^\top \nabla_{\mathbf{W}}\mathcal{L}(\mathbf{W};\mathbf{X})\mathbf{Q}_\kappa, \qquad (10)$$

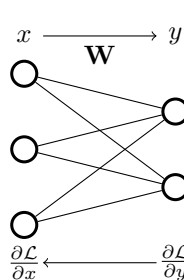 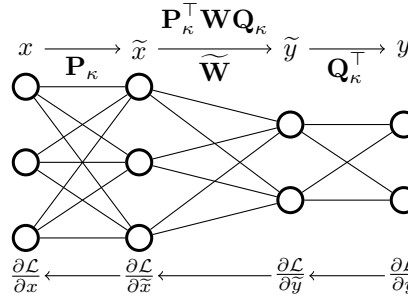

Figure 3: Illustration of network reparameterization induced by gradient projection.

which is equivalent to our *two-sided* gradient projection. Therefore, we can derive the preconditioned gradient descent for the rotated network using the following update:

$$\widetilde{\mathbf{W}}_{t+1} = \widetilde{\mathbf{W}}_{t+1} - \eta_t \frac{\widetilde{\mathbf{G}}_t}{\sqrt{\mathrm{EMA}_{\tau \in [t]} \left[ \widetilde{\mathbf{G}}_\tau^2 + \epsilon \right]}} \tag{11}$$

of which, we dropped the vectorization to ensure dimensional compatibility. Since $\mathbf{P}_\kappa$ and $\mathbf{Q}_\kappa$ are full-rank orthogonal matrices, the reparameterization is invertible and thus the update on the original parameter can be obtained by a projection back step as:

$$\mathbf{W}_{t+1} = \mathbf{W}_{t+1} - \eta_t \mathbf{P}_\kappa \frac{\widetilde{\mathbf{G}}_t}{\sqrt{\mathrm{EMA}_{\tau \in [t]} \left[ \widetilde{\mathbf{G}}_\tau^2 + \epsilon \right]}} \mathbf{Q}_\kappa^\top, \quad t \in [\kappa T, (\kappa + 1)T] \tag{12}$$

### 3.3. FINAL ALGORITHM AND RELATED WORKS

We provide the details of our proposal in Algorithm 1, which encompasses both *two-sided* and *one-sided* rotation versions. We also employ an exponential moving average of the projected gradients $\widetilde{\mathbf{G}}$ to derive the first-order moment estimation $\mathbf{M}_t$. This accumulation is performed before applying preconditioning. It should be noted that our implementation requires `torch.linalg.svd(`$\mathbf{G}_t$`, full_matrices=True)` to extract the full projection matrices $\mathbf{P}_t, \mathbf{Q}_t$ in each period. For Ada-Diag++, we recommend implementing the first SVD step at the $T$'th iteration of the training process to avoid numerical issues.

Given the flexibility of our framework, we can adapt it for other adaptive optimizers like Adafactor. In Appendix A, we provided variants of the algorithm tailored for Adafactor, along with empirical results evaluating its performance.

In connection with other existing algorithms, several prior works on optimization are relevant to our framework. George et al. (2018) and Liu et al. (2018) proposed utilizing the eigenbasis of the Fisher Information Matrix to construct diagonal preconditioning approximations within the natural gradient or online Laplace approximation families. Similarly, Anil et al. (2020) extends this idea by leveraging the eigendecomposition of Shampoo's preconditioners as a basis for the diagonal transformations.

Our method, in contrast, focuses on diagonalizing the preconditioner matrix within the generalized family of adaptive moment-based optimization algorithms, which includes Adam and Adafactor optimizers as specific cases. While primarily inspired by the critical idea of gradient projection in GaLore, we explore the full-rank projection case and thus move beyond GaLore's main focus on memory efficiency. We also acknowledge the concurrent work by SOAP (Vyas et al., 2024), which obtains the projection matrices $\mathbf{P}_t$ and $\mathbf{Q}_t$ by performing eigendecomposition on the accumulators of $\mathbf{G}_t \mathbf{G}_t^\top$ and $\mathbf{G}_t^\top \mathbf{G}_t$ (referred to as Shampoo's preconditioners), respectively. Essentially, the eigenvector matrix retrieved from the eigendecomposition of $\mathbf{G}_t \mathbf{G}_t^\top$ (and $\mathbf{G}_t^\top \mathbf{G}_t$) corresponds to the left (and right) singular matrix of $\mathbf{G}_t$. Our proposal is therefore effectively equivalent to SOAP without accumulations, resulting in enhanced memory efficiency. From a practical standpoint, our algorithms can substantially outperform Adam with only a manageable overhead.

## 4. CONVERGENCE ANALYSIS

In comparison to optimizers such as AdaGrad, Adam, and variants, the algorithm 1 introduces additional projection matrices $\{\mathbf{P}_t, \mathbf{Q}_t\}$, resulting in complex interactions between the model parameter $\mathbf{W}_t$ and optimization states $\mathbf{S}_t = \{\mathbf{M}_t, \mathbf{V}_t\}$. It is important to note that the second-order momentum $\mathbf{V}_t$ is the key factor in our framework. Without this quantity, the proposed algorithm would degenerate to the standard gradient descent with momentum, eliminating any potential improvements. This implies that the algorithms cannot be reduced to a simpler form for analyzing convergence guarantees. Moreover, applying SVD periodically to produce projection matrices also causes intricate behaviors in the dynamic subspace of optimization trajectory. It is unclear whether the momentum estimates accumulated across previous subspaces would be consistent with each other and advantageous for updates conducted in subsequent subspaces.

Inspired by the recent work on Online Subspace Descent (Liang et al., 2024), we leverage the Hamiltonian descent framework to tackle these challenges. Essentially, this framework investigates continuous-time ODE forms of optimizers in the limit of infinitesimal step size. In this setting, the optimizers will minimize an associated Hamiltonian function $\mathcal{H}(.)$, which is an augmented version of the original objective $\mathcal{L}(.)$. For example, we can derive continuous-time form for Adam optimizer as:

$$\frac{\mathrm{d}}{\mathrm{d}t}\mathbf{W}_t = -\mathbf{M}_t/(\sqrt{\mathbf{V}_t} + \epsilon), \quad \frac{\mathrm{d}}{\mathrm{d}t}\mathbf{M}_t = \mathbf{G}_t - \mathbf{M}_t, \quad \frac{\mathrm{d}}{\mathrm{d}t}\mathbf{V}_t = \mathbf{G}_t^2 - \mathbf{V}_t,$$

which yields a Hamiltonian functions defined by: $\mathcal{H}(\mathbf{W}, \mathbf{M}, \mathbf{V}) = \mathcal{L}(\mathbf{W}) + \frac{1}{2}\left\langle \mathbf{M}/(\sqrt{\mathbf{V}} + \epsilon), \mathbf{M}\right\rangle$.

**Proposition 1.** *Using this general approach, we formulate continuous-time forms for AdaDiag and AdaDiag++ as follows:*

$$(\textit{AdaDiag}): \quad \frac{\mathrm{d}}{\mathrm{d}t}\mathbf{W}_t = -\mathbf{P}_t\frac{\mathbf{M}_t}{\sqrt{\mathbf{V}_t} + \epsilon}, \quad \frac{\mathrm{d}}{\mathrm{d}t}\mathbf{M}_t = \mathbf{P}_t^\top\mathbf{G}_t - \mathbf{M}_t, \quad \frac{\mathrm{d}}{\mathrm{d}t}\mathbf{V}_t = (\mathbf{P}_t^\top\mathbf{G}_t)^2 - \mathbf{V}_t$$

$$(\textit{AdaDiag++}): \quad \frac{\mathrm{d}}{\mathrm{d}t}\mathbf{W}_t = -\mathbf{P}_t\frac{\mathbf{M}_t}{\sqrt{\mathbf{V}_t} + \epsilon}\mathbf{Q}_t^\top, \quad \frac{\mathrm{d}}{\mathrm{d}t}\mathbf{M}_t = \mathbf{P}_t^\top\mathbf{G}_t\mathbf{Q}_t - \mathbf{M}_t, \quad \frac{\mathrm{d}}{\mathrm{d}t}\mathbf{V}_t = (\mathbf{P}_t^\top\mathbf{G}_t\mathbf{Q}_t)^2 - \mathbf{V}_t$$

*Both yield the same Hamiltonian function:* $\mathcal{H}(\mathbf{W}, \mathbf{M}, \mathbf{V}) = \mathcal{L}(\mathbf{W}) + \frac{1}{2}\left\langle \mathbf{M}/(\sqrt{\mathbf{V}} + \epsilon), \mathbf{M}\right\rangle$.

**Convergence to Local Optima**. The key properties is that the function $\mathcal{H}(.)$ is monotonically non-decreasing along its ODE trajectory, namely $\frac{\mathrm{d}}{\mathrm{d}t}\mathcal{H}(\mathbf{W}_t, \mathbf{S}_t) \leq 0, \forall t$. By LaSalle's Invariance principle, the set of accumulation points $(\mathbf{W}_t, \mathbf{S}_t)$ must be contained in $\mathcal{I}$, where $\mathcal{I} = \{\text{the union of complete trajectories satisfying } \frac{\mathrm{d}}{\mathrm{d}t}\mathcal{H}(\mathbf{W}_t, \mathbf{S}_t) = 0\}$. The points in limit set $\mathcal{I}$ should satisfy $\mathbf{P}_t^\top\nabla\mathcal{L}(\mathbf{W}_t) \equiv 0$ for AdaDiag or $\mathbf{P}_t^\top\nabla\mathcal{L}(\mathbf{W}_t)\mathbf{Q}_t \equiv 0$ for AdaDiag++, respectively. Since $\mathbf{P}_t, \mathbf{Q}_t$ are full-rank orthogonal matrices, we must have $\nabla\mathcal{L}(\mathbf{W}_t) \equiv 0$, which indicates that all trajectories will converge to local optimal points. Detailed analysis is provided in Appendix B.

## 5. EXPERIMENTS

In this section, we conduct several experiments on image classification and language modeling tasks to verify the efficiency of our algorithm. We will also demonstrate that our general framework can be effectively applied to enhance adaptive moment-based optimizers such as AdamW and Adafactor.

### 5.1. IMAGE CLASSIFICATION

We first evaluated the optimization algorithms, including AdamW, AdaDiag, and AdaDiag++, by pretraining the ImageNet1k dataset from scratch using ResNets and Vision Transformers (ViTs) architectures. The images underwent Inception-style cropping (Szegedy et al., 2016) and random horizontal flipping during pre-processing. We trained ResNet50 for 90 epochs with a batch size of 1024, utilizing a cosine learning rate decay scheduler. For ViTs, we conducted training over 300 epochs with a batch size of 4096, using a learning rate schedule that included a 10,000-step

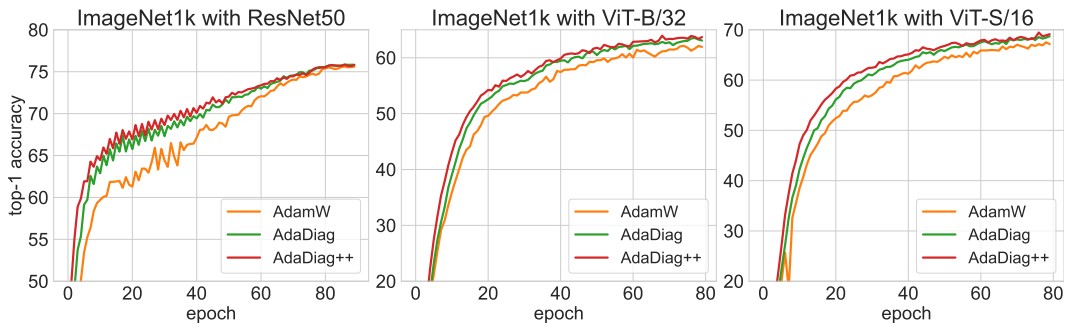

Figure 4: Top-1 Accuracy of optimizers in pretraining ResNet50, ViT-B/32, and ViT-S/16 from scratch on the ImageNet1k. For better ViT's visualization, we crop the learning curve up to epoch 80.

Table 1: Comparison of Adam and AdaDiag on pre-training ResNets and ViTs architectures with ImageNet1k dataset. Top-1 accuracy on the validation set is reported.

| Models | ResNet50 | ViT-S/16 | ViT-B/32 |
|---|---|---|---|
| Adam (Kingma & Ba, 2014) | 75.61 | 78.35 | 72.20 |
| AdaDiag (ours) | 75.85 | **79.18** | **73.39** |
| AdaDiag++ (ours) | **75.86** | 79.14 | 73.24 |

warmup followed by linear decay. Additionally, we employed strong data augmentations, such as RandAugment (2,15) (Cubuk et al., 2019) and mixup (0.5) (Zhang et al., 2017), to further improve the performance of the ViTs. For hyperparameters settings, such as learning rate ($lr$), weight decay ($\lambda$), and dropout rate ($dr$), we opted for recommended configurations from prior research and left them in Appendix C.

As shown in Figure 4, AdaDiag and AdaDiag++ exhibit substantial improvements in convergence speed compared to the baseline AdamW. For the ViT-B/32 and ViT-S/16 models, we focus on the first third of the training phase so that we can observe notable accelerations across the three models. For the full performances, we refer to the results in Figure 8 in Appendix D. The final results at convergence are provided in Table 1. By more accurate approximations of the preconditioner matrix, we expect that AdaDiag++ can navigate the complex curvature more efficiently and thus provide better convergence properties, even compared to AdaDiag. The results appear to support this argument. It's important to mention that these two algorithms would perform similarly after converging at some point. We hypothesize that the optimization trajectory will eventually converge to a stable region where the gradients reside in a very low-dimensional subspace. The precondition approximations, at this stage, become less critical as the optimization process focuses on fine-tuning within this reduced space.

## 5.2. LANGUAGE MODELING

We apply the algorithms to pre-train LLaMA-based large language model, with RMSNorm and SwiGLU activations (Zhang & Sennrich, 2019; Shazeer, 2020; Touvron et al., 2023), on the C4 dataset (Raffel et al., 2020). We measured the perplexity of the models on the validation set throughout training to assess convergence properties and final model performance. Specifically, we trained LLaMA models of sizes 60M, 130M, and 350M for 10K, 20K, and 60K steps, respectively. The learning rate schedule included a warmup phase during the first 10% of the training steps, followed by cosine annealing that decayed the learning rate to 10% of its initial value. All models used a maximum sequence length of 256 and a batch size of 512.

The results are provided in Figure 5 and Table 2. Our optimizer AdaDiag consistently outperforms Adam on various sizes of LLaMA models. In particular, AdaDiag can achieve 1.8x-2x speed-up compared to Adam, achieving the same perplexity with half fewer steps. Because of resource constraints, we were unable to conduct experiments with billion-parameter models. However, we are confident that similar results can be reliably achieved with larger-scale models.

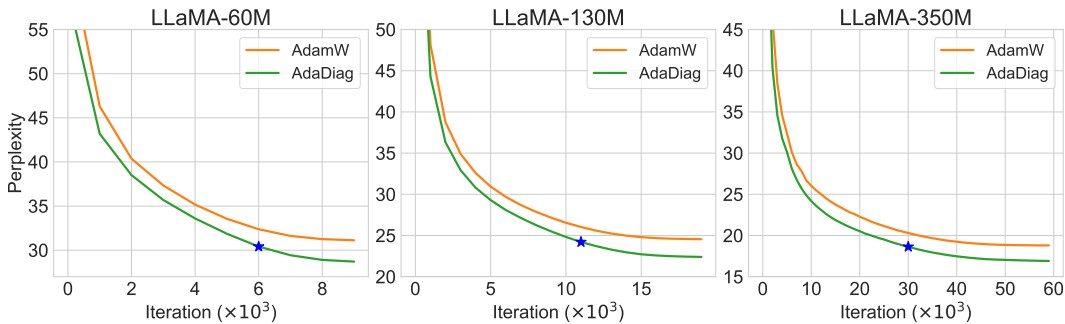

Figure 5: Training progression for pre-training LLaMA models on C4 dataset.

Table 2: Comparison of Adam and AdaDiag on pre-training LLaMA models with the C4 dataset. Validation perplexity is reported for models with 60/130/350 Million parameters trained for 1.1/2.2/6.4 Billion training tokens.

| Optimizer | Validation Perplexity | | |
|---|---|---|---|
| | 60M / 1.1B | 130M / 2.2B | 350M / 6.4B |
| Adam (Kingma & Ba, 2014) | 31.12 | 24.55 | 18.79 |
| AdaDiag (ours) | **28.71** | **22.40** | **16.91** |

### 5.3. IMPROVING COMPUTATIONAL AND MEMORY EFFICIENCY

Although our proposal can greatly enhance the performance, it comes with an inevitable trade-off in algorithmic complexity. While the total computational overhead induced by periodic SVD is negligible (less than $10\%$), the memory usage caused by the full-rank projection may limit the applicability of the algorithms. To address this challenge, we propose to apply the general framework to memory-efficiency optimization methods such as Adafactor (Shazeer & Stern, 2018). This optimizer employs a rank-1 parameterization to the second-order moment and thus offers a sublinear memory cost. In Table 3, we provide an analysis of the complexity of these algorithms.

Table 3: Memory requirements for different optimizers, with weight parameter of size $m \times n, m \le n$.

| Optimizers | Adam | AdaDiag | AdaDiag++ | Adafactor | | AdafacDiag | |
|---|---|---|---|---|---|---|---|
| | | | | w/ m. | w/o m. | w/ m. | w/o m. |
| Weights | $mn$ | $mn$ | $mn$ | $mn$ | $mn$ | $mn$ | $mn$ |
| Gradient | $mn$ | $mn$ | $mn$ | $mn$ | $mn$ | $mn$ | $mn$ |
| Optim. States | $2mn$ | $m^2 + 2mn$ | $(m+n)^2$ | $mn + m + n$ | $m + n$ | $m^2 + mn + m + n$ | $m^2 + m + n$ |

Adafactor with momentum (w/ m.) has demonstrated results comparable to Adam on various tasks (Shazeer & Stern, 2018; Chen et al., 2024). Hence, the AdafacDiag integration could potentially surpass Adam's performance while maintaining a similar complexity. We conducted several experiments to validate this hypothesis, with the results presented in Appendix A.

## 6. DISCUSSION

In this work, we proposed an efficient approach to improve adaptive moment-based optimizers by introducing a preconditioner diagonalization strategy. By leveraging an invertible transformation, we were able to enhance the reliability of the diagonal approximation used for the preconditioner matrix, resulting in a more effective estimation of second-order statistics. Our empirical evaluations demonstrated significant improvements in both convergence speed and final model performance across several standard tasks. Furthermore, our work also underscores the significance of employing structural preconditioning techniques to improve existing adaptive learning rate optimizers. Devising new preconditioners or exploring network reparameterization present promising approaches to this

problem. In addition, it is vital to establish theoretically sound frameworks to better understand adaptive moment-based optimizers in dynamic subspaces. This would enable us to identify the optimal subspace where the moment estimates can be applied most effectively.

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

## A.  ADAFACTOR OPTIMIZER

---

**Algorithm 2** Adafactor for matrix parameter $\mathbf{W}$ of size $m \times n$, $m \leq n$.

---

**Inputs:** moment decay coefficients $\beta_1, \beta_2$, smoothing term $\epsilon = 10^{-30}$, regularization constant $\lambda$
**Initialization:** weight parameters $\mathbf{W}_0 \in \mathbb{R}^{m \times n}$, initial moments $\mathbf{M}_0, \mathbf{r}_0, \mathbf{s}_0 \leftarrow 0$
**repeat**
    $t \leftarrow t + 1$
    $\mathbf{G}_t = \nabla \mathcal{L}(\mathbf{W}_t; \mathcal{B}_t)$
    $\mathbf{r}_t = \hat{\beta}_{2t} \mathbf{r}_{t-1} + (1 - \hat{\beta}_{2t}) \left[ (\mathbf{G}_t)^2 + \epsilon \right] \mathbf{1}_n$
    $\mathbf{s}_t = \hat{\beta}_{2t} \mathbf{s}_{t-1} + (1 - \hat{\beta}_{2t}) \left[ (\mathbf{G}_t^\top)^2 + \epsilon \right] \mathbf{1}_m$
    $\mathbf{V}_t = \mathbf{r}_t \mathbf{s}_t^\top / (\mathbf{1}_m^\top \mathbf{r}_t)$
    $\mathbf{M}_t = \hat{\beta}_{1t} \mathbf{M}_{t-1} + (1 - \hat{\beta}_{1t}) \text{clip} \left( \mathbf{G}_t / \left( \sqrt{\mathbf{V}_t} \right) \right)$
    $\mathbf{W}_{t+1} = \mathbf{W}_t - \eta_t \left( \mathbf{M}_t + \lambda \mathbf{W}_t \right)$
**until** *stopping criterion is met*
**return** optimized parameter $\mathbf{W}_t$

---

**Algorithm 3** AdafacDiag for matrix parameter $\mathbf{W}$ of size $m \times n$, $m \leq n$.

---

**Inputs:** moment decay coefficients $\beta_1, \beta_2$, smoothing term $\epsilon = 10^{-30}$, regularization constant $\lambda$
**Initialization:** weight parameters $\mathbf{W}_0 \in \mathbb{R}^{m \times n}$, initial moments $\mathbf{M}_0, \mathbf{r}_0, \mathbf{s}_0 \leftarrow 0$
**repeat**
    $t \leftarrow t + 1$
    $\mathbf{G}_t = \nabla \mathcal{L}(\mathbf{W}_t; \mathcal{B}_t)$
    **if** $t \bmod T = 0$ **then**
        $\mathbf{P}_t, \_, \mathbf{Q}_t^\top = \texttt{torch.linalg.svd}(\mathbf{G}_t, \texttt{full\_matrices=True})$
    **else**
        $\mathbf{P}_t, \mathbf{Q}_t^\top \leftarrow \mathbf{P}_{t-1}, \mathbf{Q}_{t-1}^\top$
    **end if**
    $\widetilde{\mathbf{G}}_t = \mathbf{P}_t^\top \mathbf{G}_t$
    $\mathbf{r}_t = \hat{\beta}_{2t} \mathbf{r}_{t-1} + (1 - \hat{\beta}_{2t}) \left[ (\widetilde{\mathbf{G}}_t)^2 + \epsilon \right] \mathbf{1}_n$
    $\mathbf{s}_t = \hat{\beta}_{2t} \mathbf{s}_{t-1} + (1 - \hat{\beta}_{2t}) \left[ (\widetilde{\mathbf{G}}_t^\top)^2 + \epsilon \right] \mathbf{1}_m$
    $\mathbf{V}_t = \mathbf{r}_t \mathbf{s}_t^\top / (\mathbf{1}_m^\top \mathbf{r}_t)$
    $\mathbf{M}_t = \hat{\beta}_{1t} \mathbf{M}_{t-1} + (1 - \hat{\beta}_{1t}) \text{clip} \left( \widetilde{\mathbf{G}}_t / \left( \sqrt{\mathbf{V}_t} \right) \right)$
    $\mathbf{W}_{t+1} = \mathbf{W}_t - \eta_t \left( \mathbf{P}_t \mathbf{M}_t + \lambda \mathbf{W}_t \right)$
**until** *stopping criterion is met*
**return** optimized parameter $\mathbf{W}_t$

---

Adafactor (Shazeer & Stern, 2018) proposed an efficient rank-1 parameterization for the second-order momentum $\mathbf{V}$, which is widely adopted in adaptive optimization methods like RMSprop, Adam, and its variants. The factorization was derived by minimizing the total elementwise I-divergence subject to componentwise non-negative constraints:

$$\underset{\mathbf{r} \in \mathbb{R}^m, \mathbf{s} \in \mathbb{R}^n}{\text{minimize}} \sum_{i=1}^{m} \sum_{j=1}^{n} d(V_{ij}, r_i s_j)$$

in which $r_i \geq 0, s_j \geq 0$ and $d(p, q) = p \log \frac{p}{q} - p + q$.

Solving this problem results in a closed-form solution denoted by $\mathbf{r} = \mathbf{V} \mathbf{1}_n, \mathbf{s} = \mathbf{V}^\top \mathbf{1}_m / \mathbf{r}^\top \mathbf{1}_n$. Intuitively, Adafactor tracks the moving averages of the row and column sums of squared gradients throughout iterations, yielding factored second-moment estimators $\mathbf{r}_t$ and $\mathbf{s}_t$. It then reconstructs a low-rank parameterization of the second-order momentum using a normalized outer product $\mathbf{r}_t \mathbf{s}_t^\top / (\mathbf{1}_m^\top \mathbf{r}_t)$. This method is computationally efficient and scalable, as it directly offers analytical formulations without requiring further approximations.

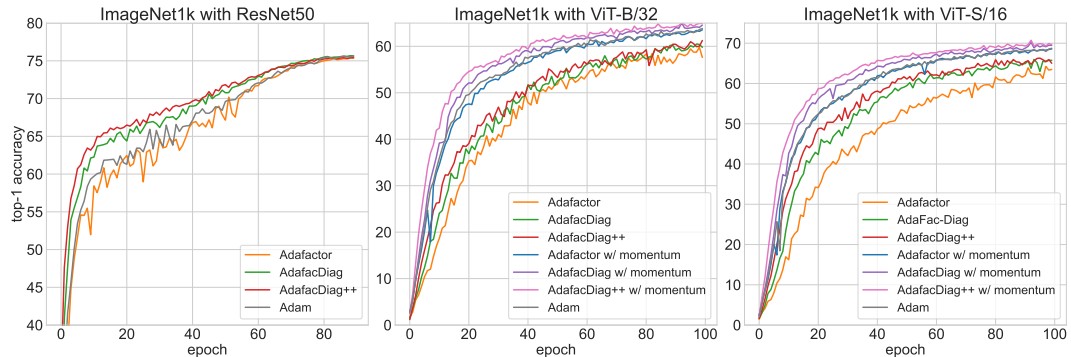

Figure 6: Top-1 Accuracy of optimizers in pre-training ResNet50, ViT-B/32, and ViT-S/16 from scratch on the ImageNet1k.

Table 4: Comparison of Adam, Adafactor, and AdafacDiag on pre-training ResNets and ViTs architectures with ImageNet1k dataset. Top-1 accuracy on the validation set is reported.

| Models | ResNet50 | ViT-S/16 | ViT-B/32 |
|---|---|---|---|
| AdamW | 75.61 | 78.35 | 72.20 |
| Adafactor | 75.37 | 77.14 | 71.42 |
| AdafacDiag | **75.68** | 78.45 | 72.54 |
| AdafacDiag++ | 75.60 | 78.56 | 72.78 |
| Adafactor w/ momentum | - | 78.44 | 72.31 |
| AdafacDiag w/ momentum | - | **78.90** | **73.24** |
| AdafacDiag++ w/ momentum | - | 78.66 | **73.28** |

Incorporating Adafactor into our framework offers significant computational and memory efficiency benefits. This can be evident in Table 3, from which AdafacDiag demonstrates lower complexity in optimization states when compared to Adam. To further assess the effectiveness of this integration, we carried out experiments similar to those described in the main text. As shown in Figure 6, AdafacDiag (on ResNet50) and AdafacDiag with momentum (on ViTs) can outperform Adam by noticeable margins. The experiments on LLaMA-based models using the C4 dataset also delivered consistent results, presented in Figure 7 and Table 5. These advantages highlight the potential of utilizing these algorithms in a wide range of real-world tasks, particularly in large-scale applications.

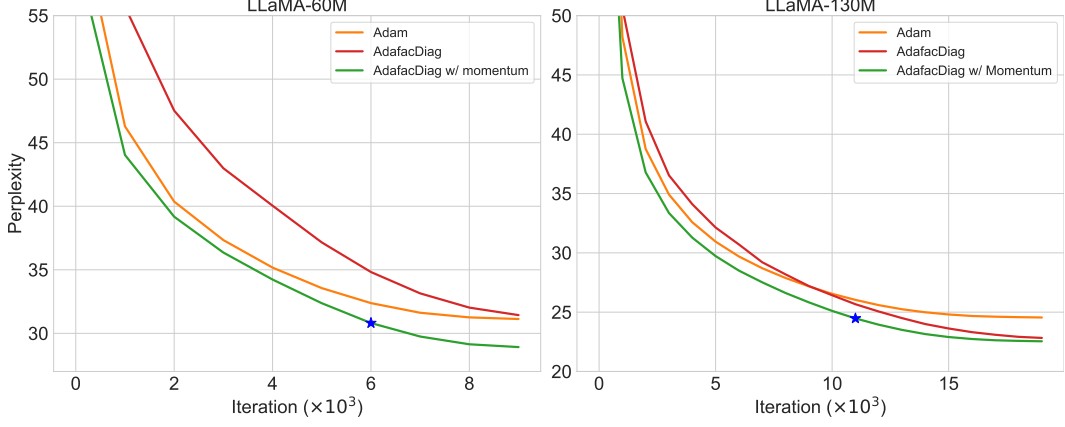

Figure 7: Training progression of Adam and AdafacDiag for pre-training LLaMA models on C4 dataset.

Table 5: Comparison of Adam and AdafacDiag on pre-training LLaMA models with the C4 dataset. Validation perplexity is reported for models with 60/130 Million parameters trained for 1.1/2.2 Billion training tokens.

| Optimizer | Validation Perplexity | |
|---|---|---|
| | 60M / 1.1B | 130M / 2.2B |
| Adam (Kingma & Ba, 2014) | 31.12 | 24.55 |
| AdafacDiag (ours) | 31.43 | **22.82** |
| AdafacDiag w/ momentum (ours) | **28.91** | **22.54** |

## B. HAMILTONIAN FUNCTION ANALYSES

We first formalize a unified view of adaptive moment-based optimizers in the update scheme as:

$$\mathbf{W}_{t+1} = \mathbf{W}_t + \phi_t(\mathbf{S}_t) \qquad\qquad \mathbf{S}_t = \psi_t(\mathbf{S}_{t-1}, \nabla\mathcal{L}(\mathbf{W}_t)) \qquad (13)$$

where $\mathbf{S}_t$ is the optimization state, and $\phi_t, \psi_t$ are some mapping functions. One powerful approach to studying the dynamic behavior of these optimizers is to examine their continuous-time ODE forms in the limit of infinitesimal step size (Maddison et al., 2018; Gao et al., 2022; Chen et al., 2023). It provides insights into the asymptotic convergence of the algorithms, abstracting away the choices of step size, discretization, and accumulation errors.

We observe that the update 13 can be discretized from the following continuous-time form:

$$\frac{\mathrm{d}}{\mathrm{d}t}\mathbf{W}_t = \partial_{\mathbf{S}}\mathcal{H}(\mathbf{W}_t, \mathbf{S}_t) - \Phi(\partial_{\mathbf{W}}\mathcal{H}(\mathbf{W}_t, \mathbf{S}_t))$$
$$\frac{\mathrm{d}}{\mathrm{d}t}\mathbf{S}_t = -\partial_{\mathbf{W}}\mathcal{H}(\mathbf{W}_t, \mathbf{S}_t) - \Psi(\partial_{\mathbf{S}}\mathcal{H}(\mathbf{W}_t, \mathbf{S}_t)) \qquad (14)$$

where $\mathcal{H}(.)$ is a Hamiltonian function that satisfies:

$$\min_{\mathbf{S}} \mathcal{H}(\mathbf{W}, \mathbf{S}) = \mathcal{L}(\mathbf{W}) \quad \forall\mathbf{W},$$

meaning that minimizing $\mathcal{H}(\mathbf{W}, \mathbf{S})$ will reduce to minimizing the original objective $\mathcal{L}(\mathbf{W})$. Additionally, $\Phi, \Psi$ are two monotonic mapping satisfying:

$$\|\mathbf{A}\|_{\Phi}^2 = \langle \mathbf{A}, \Phi(\mathbf{A}) \rangle \geq 0, \qquad\qquad \|\mathbf{A}\|_{\Psi}^2 = \langle \mathbf{A}, \Psi(\mathbf{A}) \rangle \geq 0, \quad \forall\mathbf{A}$$

The key properties is that $\mathcal{H}(\mathbf{W}, \mathbf{S})$ is monotonically non-decreasing along the trajectory 14:

$$\frac{\mathrm{d}}{\mathrm{d}t}\mathcal{H}(\mathbf{W}_t, \mathbf{S}_t)$$
$$= \left\langle \partial_{\mathbf{W}}\mathcal{H}(\mathbf{W}_t, \mathbf{S}_t), \frac{\mathrm{d}}{\mathrm{d}t}\mathbf{W}_t \right\rangle + \left\langle \partial_{\mathbf{S}}\mathcal{H}(\mathbf{W}_t, \mathbf{S}_t), \frac{\mathrm{d}}{\mathrm{d}t}\mathbf{S}_t \right\rangle$$
$$= -\langle \partial_{\mathbf{W}}\mathcal{H}(\mathbf{W}_t, \mathbf{S}_t), \Phi(\partial_{\mathbf{W}}\mathcal{H}(\mathbf{W}_t, \mathbf{S}_t)) \rangle - \langle \partial_{\mathbf{S}}\mathcal{H}(\mathbf{W}_t, \mathbf{S}_t), \Psi(\partial_{\mathbf{S}}\mathcal{H}(\mathbf{W}_t, \mathbf{S}_t)) \rangle$$
$$= -\|\partial_{\mathbf{W}}\mathcal{H}(\mathbf{W}_t, \mathbf{S}_t)\|_{\Phi}^2 - \|\partial_{\mathbf{S}}\mathcal{H}(\mathbf{W}_t, \mathbf{S}_t)\|_{\Psi}^2 \leq 0,$$

where we cancel out the cross terms $\langle \partial_{\mathbf{W}}\mathcal{H}(\mathbf{W}_t, \mathbf{S}_t), \partial_{\mathbf{S}}\mathcal{H}(\mathbf{W}_t, \mathbf{S}_t) \rangle$ to get the second equation.

Let's take Adam optimizer as a specific example, we have its ODE form represented as:

$$\frac{\mathrm{d}}{\mathrm{d}t}\mathbf{W}_t = -\frac{\mathbf{M}_t}{\sqrt{\mathbf{V}_t} + \epsilon}, \quad \frac{\mathrm{d}}{\mathrm{d}t}\mathbf{M}_t = \mathbf{G}_t - \mathbf{M}_t, \quad \frac{\mathrm{d}}{\mathrm{d}t}\mathbf{V}_t = \mathbf{G}_t^2 - \mathbf{V}_t, \qquad (15)$$

of which the Hamiltonian function is defined by:

$$\mathcal{H}(\mathbf{W}, \mathbf{M}, \mathbf{V}) = \mathcal{L}(\mathbf{W}) + \frac{1}{2}\left\langle \frac{\mathbf{M}}{\sqrt{\mathbf{V}} + \epsilon}, \mathbf{M} \right\rangle \qquad (16)$$

We have the derivative:

$$\frac{\mathrm{d}}{\mathrm{d}t}\mathcal{H}(\mathbf{W}_t, \mathbf{M}_t, \mathbf{V}_t)$$

$$= \left\langle \mathbf{G}_t, \frac{\mathrm{d}}{\mathrm{d}t}\mathbf{W}_t \right\rangle + \left\langle \frac{\mathbf{M}_t}{\sqrt{\mathbf{V}_t}+\epsilon}, \frac{\mathrm{d}}{\mathrm{d}t}\mathbf{M}_t \right\rangle - \frac{1}{4}\left\langle \frac{\mathbf{M}_t^2}{\sqrt{\mathbf{V}_t}\odot\left(\sqrt{\mathbf{V}_t}+\epsilon\right)^2}, \frac{\mathrm{d}}{\mathrm{d}t}\mathbf{V}_t \right\rangle$$

$$= \left\langle \mathbf{G}_t, -\frac{\mathbf{M}_t}{\sqrt{\mathbf{V}_t}+\epsilon} \right\rangle + \left\langle \frac{\mathbf{M}_t}{\sqrt{\mathbf{V}_t}+\epsilon}, \mathbf{G}_t - \mathbf{M}_t \right\rangle - \frac{1}{4}\left\langle \frac{\mathbf{M}_t^2}{\sqrt{\mathbf{V}_t}\odot\left(\sqrt{\mathbf{V}_t}+\epsilon\right)^2}, \mathbf{G}_t^2 - \mathbf{V}_t \right\rangle$$

$$= -\left\langle \frac{\mathbf{M}_t}{\sqrt{\mathbf{V}_t}+\epsilon}, \mathbf{M}_t \right\rangle + \frac{1}{4}\left\langle \frac{\mathbf{M}_t^2}{\left(\sqrt{\mathbf{V}_t}+\epsilon\right)^2}, \sqrt{\mathbf{V}_t} \right\rangle - \frac{1}{4}\left\langle \frac{\mathbf{M}_t^2}{\sqrt{\mathbf{V}_t}\odot\left(\sqrt{\mathbf{V}_t}+\epsilon\right)^2}, \mathbf{G}_t^2 \right\rangle$$

$$= -\left\langle \frac{\mathbf{M}_t}{\sqrt{\mathbf{V}_t}+\epsilon}, \mathbf{M}_t \right\rangle + \frac{1}{4}\left\langle \frac{\mathbf{M}_t}{\sqrt{\mathbf{V}_t}+\epsilon}\odot\frac{\sqrt{\mathbf{V}_t}}{\sqrt{\mathbf{V}_t}+\epsilon}, \mathbf{M}_t \right\rangle - \frac{1}{4}\left\langle \frac{\mathbf{M}_t^2}{\sqrt{\mathbf{V}_t}\odot\left(\sqrt{\mathbf{V}_t}+\epsilon\right)^2}, \mathbf{G}_t^2 \right\rangle$$

$$\leq -\frac{1}{4}\left\langle \frac{\mathbf{M}_t^2}{\sqrt{\mathbf{V}_t}\odot\left(\sqrt{\mathbf{V}_t}+\epsilon\right)^2}, \mathbf{G}_t^2 \right\rangle \leq 0.$$

A similar analysis can be applied to the AdaDiag and AdaDiag++. In AdaDiag's case, we have the continuous form:

$$\frac{\mathrm{d}}{\mathrm{d}t}\mathbf{W}_t = -\mathbf{P}_t\frac{\mathbf{M}_t}{\sqrt{\mathbf{V}_t}+\epsilon}, \quad \frac{\mathrm{d}}{\mathrm{d}t}\mathbf{M}_t = \mathbf{P}_t^\top\mathbf{G}_t - \mathbf{M}_t, \quad \frac{\mathrm{d}}{\mathrm{d}t}\mathbf{V}_t = (\mathbf{P}_t^\top\mathbf{G}_t)^2 - \mathbf{V}_t, \quad (17)$$

yielding the Hamiltonian function $\mathcal{H}(\mathbf{W}, \mathbf{M}, \mathbf{V}) = \mathcal{L}(\mathbf{W}) + \frac{1}{2}\left\langle \frac{\mathbf{M}}{\sqrt{\mathbf{V}}+\epsilon}, \mathbf{M} \right\rangle$, for which:

$$\frac{\mathrm{d}}{\mathrm{d}t}\mathcal{H}(\mathbf{W}_t, \mathbf{M}_t, \mathbf{V}_t)$$

$$= \left\langle \mathbf{G}_t, \frac{\mathrm{d}}{\mathrm{d}t}\mathbf{W}_t \right\rangle + \left\langle \frac{\mathbf{M}_t}{\sqrt{\mathbf{V}_t}+\epsilon}, \frac{\mathrm{d}}{\mathrm{d}t}\mathbf{M}_t \right\rangle - \frac{1}{4}\left\langle \frac{\mathbf{M}_t^2}{\sqrt{\mathbf{V}_t}\odot\left(\sqrt{\mathbf{V}_t}+\epsilon\right)^2}, \frac{\mathrm{d}}{\mathrm{d}t}\mathbf{V}_t \right\rangle$$

$$= \left\langle \mathbf{G}_t, -\mathbf{P}_t\frac{\mathbf{M}_t}{\sqrt{\mathbf{V}_t}+\epsilon} \right\rangle + \left\langle \frac{\mathbf{M}_t}{\sqrt{\mathbf{V}_t}+\epsilon}, \mathbf{P}_t^\top\mathbf{G}_t - \mathbf{M}_t \right\rangle - \frac{1}{4}\left\langle \frac{\mathbf{M}_t^2}{\sqrt{\mathbf{V}_t}\odot\left(\sqrt{\mathbf{V}_t}+\epsilon\right)^2}, (\mathbf{P}_t^\top\mathbf{G}_t)^2 - \mathbf{V}_t \right\rangle$$

$$= -\left\langle \frac{\mathbf{M}_t}{\sqrt{\mathbf{V}_t}+\epsilon}, \mathbf{M}_t \right\rangle + \frac{1}{4}\left\langle \frac{\mathbf{M}_t^2}{\left(\sqrt{\mathbf{V}_t}+\epsilon\right)^2}, \sqrt{\mathbf{V}_t} \right\rangle - \frac{1}{4}\left\langle \frac{\mathbf{M}_t^2}{\sqrt{\mathbf{V}_t}\odot\left(\sqrt{\mathbf{V}_t}+\epsilon\right)^2}, (\mathbf{P}_t^\top\mathbf{G}_t)^2 \right\rangle$$

$$\leq -\frac{1}{4}\left\langle \frac{\mathbf{M}_t^2}{\sqrt{\mathbf{V}_t}\odot\left(\sqrt{\mathbf{V}_t}+\epsilon\right)^2}, (\mathbf{P}_t^\top\mathbf{G}_t)^2 \right\rangle \leq 0.$$

This result implies that the points in the limit set:

$$\mathcal{I} = \{\text{the union of complete trajectories satisfying } \frac{\mathrm{d}}{\mathrm{d}t}\mathcal{H}(\mathbf{W}_t, \mathbf{M}_t, \mathbf{V}_t) = 0\}$$

must satisfy $\mathbf{P}_t^\top\mathbf{G}_t \equiv 0$. Since $\mathbf{P}_t$ is a full-rank orthogonal matrix, we have $\mathbf{G}_t \equiv 0$, indicating that the optimization algorithm converges to a local optimum.

## C. HYPERPARAMETER SETTINGS

For all optimizers, we used decay moment coefficients $(\beta_1, \beta_2) = (0.9, 0.999)$ along with the bias-correction steps, the smoothing constant $\epsilon = 10^{-8}$. Specifically with AdaDiag, AdaDiag++, we use the SVD frequency $T = 500, 200$ for image and language tasks, respectively.

For image classification, we used $(lr, \lambda) = (0.001, 0.1)$ and $(0.0003, 0.1)$ for all ResNets and ViTs experiments, respectively.

For language modeling, we tuned the learning rate over the set $\{0.003, 0.001, 0.0003, 0.0001\}$ and selected the optimal value based on validation perplexity. The specific settings are summarized in the Table 6.

Table 6: Training configuration for LLaMA models.

| LLaMA models | Tokens | Training Steps | Warmup Steps | Learning Rate |
|---|---|---|---|---|
| 60M | 1.3B | 10K | 1K | 0.003 |
| 130M | 2.6B | 20K | 2K | 0.001 |
| 350M | 7.8B | 60K | 6K | 0.001 |

## D. ADDITIONAL FIGURES

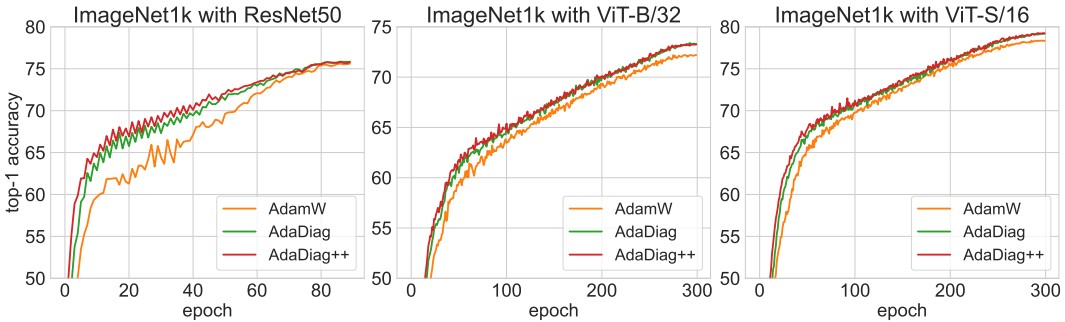

Figure 8: Top-1 Accuracy of optimizers in pre-training (to the end) ResNet50, ViT-B/32, and ViT-S/16 from scratch on the ImageNet1k.

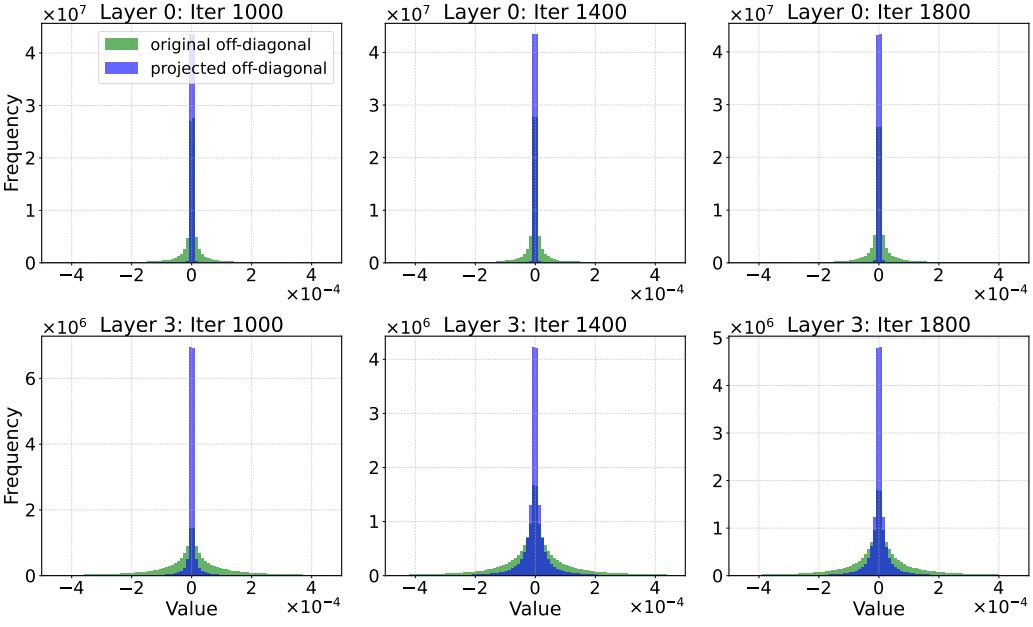

Figure 9: Histograms of off-diagonal elements $\mathcal{C}(\mathbf{G}_\tau)$ (original) and $\mathcal{C}(\widetilde{\mathbf{G}}_\tau)$ (*one-sided* projection, AdaDiag), corresponding to the two first layers of ResNet50 trained on ImageNet1k. In this experiment, we set the frequency $T = 500$ and plot histograms at iterations with and without SVD applied.

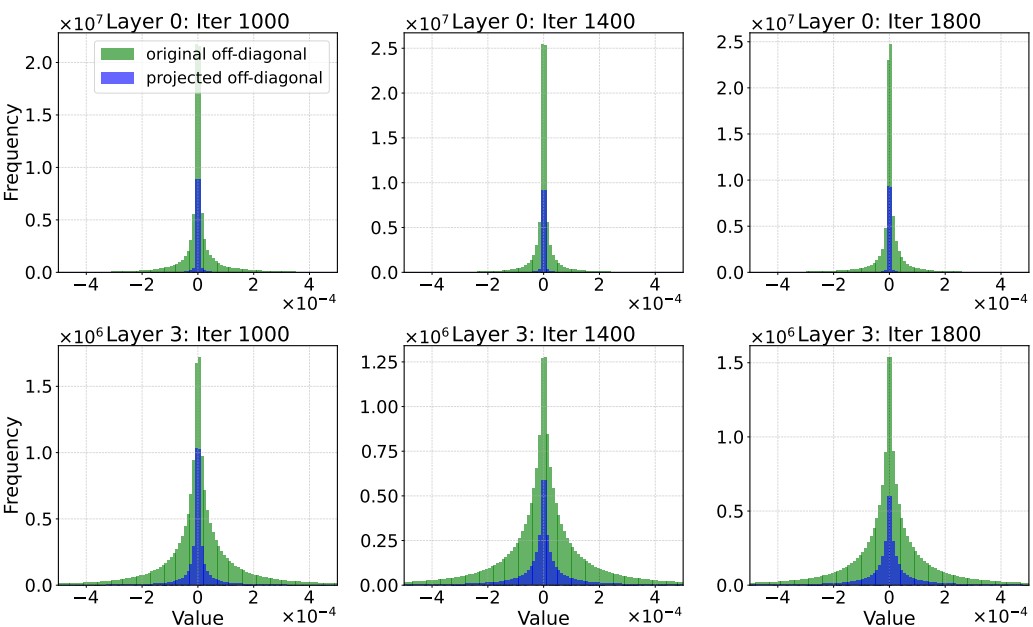

Figure 10: Histograms of off-diagonal elements $\mathcal{C}(\mathbf{G}_\tau)$ (original) and $\mathcal{C}(\widetilde{\mathbf{G}}_\tau)$ (**GaLore**), corresponding to the two first layers of ResNet50 trained on ImageNet1k. Compared to the full-rank case, the low-rank GaLore projection $(r = \min\{m, n\}/2)$ does not exhibit a discernible sparsity structure in the off-diagonal elements. GaLore even creates off-diagonal elements with larger magnitudes. Note that the support was truncated for better visualization.

