# OpenReview forum: "Improving Adaptive Moment Optimization via Preconditioner Diagonalization"
_ICLR.cc/2025/Conference — Submitted to ICLR 2025_

### Official Review · Reviewer_n6yE · 2024-11-02

**Soundness:** 2
**Presentation:** 3
**Contribution:** 2
**Rating:** 3
**Confidence:** 4

**Summary:**

This paper proposes a new optimizer with the aim of mapping the gradients in a space where the gradients are approximately diagonal and running Adam/Adafactor in that space. They show empirical improvements on language model training.

**Strengths:**

The most interesting aspects of the paper were Fig. 1, 2 and 9 which showed the off-diagonal/sparsity behavior of projected gradients.

**Weaknesses:**

I think there are two major issues with this work:

1. On the theoretical side, the line “Since $\Sigma_{\tau}$ is a diagonal matrix, we have $\text{vec}(\Sigma_{\tau}) \, \text{vec}(\Sigma_{\tau})^{\top}$ is almost diagonal (off-diagonal elements are mostly zero).” is the crux of their proof. But this is clearly not fully correct since  $\text{vec}(\Sigma_{\tau}) \, \text{vec}(\Sigma_{\tau})^{\top}$ is a rank-1 matrix. The only matrix which is both rank 1 and diagonal is a matrix with a single non-zeros entry on its diagonal.  I agree with the authors that this matrix is more diagonal than the original matrix due to $(\Sigma_{\tau}$ having a strong decay along the diagonal.

    There is in fact a bigger issue with this approach: the matrix the authors begin with $C(G_{\tau}) = \text{vec}(G_{\tau}) \, \text{vec}(G_{\tau})^{\top}$ is itself is a rank 1 matrix. There is a trivial way to rotate this matrix to make it exactly diagonal which is to use a basis with one of the elements being $\text{vec}(G_{\tau})$. If the authors believe diagonalization is the main motivation they should use any such basis. But under this basis applyling Adam on the gradient (assuming it remains nearly constant which while not true is anyways assumed by the authors since frequency > 1) will just lead to gradient normalization per layer since all of the projected gradient will lie in the coordinate corresponding to the $\text{vec}(G_{\tau})$ basis element.
2. One the empirical side:

     A. The authors do not give details of their hyperaprameters ablations for their results.

     B. Since the optimizer has connections to Shampoo the authors should compare it to Shampoo (see https://github.com/facebookresearch/optimizers/tree/main/distributed_shampoo for an implementation).


There are some other issues with the discussion of prior and related works which I can expand on once the authors address these two points.

**Questions:**

Already asked in the previous parts of the review.

---

> ### Author Response · Authors · 2024-11-25
>
> We thank the reviewer for the feedback and comments.
>
> We agree that the "diagonalization" presented in the paper is not a rigorous algebraic concept. However, we have clarified its intended meaning throughout the manuscript, describing it as a method to make the preconditioner matrix approximately diagonal.
>
> For the empirical evaluation, we will incorporate a comparison with Shampoo as suggested to strengthen the paper. Additionally, detailed hyperparameter settings are already provided in Appendix C.

---

> > ### Comment · Reviewer_n6yE · 2024-11-29
> >
> > Thank you for the response. Given my concerns, I will maintain my score.

---

### Official Review · Reviewer_SM6r · 2024-11-03

**Soundness:** 3
**Presentation:** 3
**Contribution:** 2
**Rating:** 3
**Confidence:** 4

**Summary:**

The paper proposes the gradient SVD basis for diagonalizing the preconditioner proposed by Adagrad, and then uses the standard first order algorithms such as Adam and Adafactor within the basis. It shows improved results over Adam on image and language modeling tasks.

**Strengths:**

The paper combines the notion of preconditioner diagonalization with adaptive first order methods and shows improved results on image and language modeling tasks.

**Weaknesses:**

1. Although the method proposed by the paper clearly lies within the class of second order methods, no second order baselines have been provided in the work. The paper needs to compare to other second order methods such as Shampoo, KFAC, SOAP as baselines.

2. Also, in section 3.3, the paper makes the claim that the preconditioner basis provided by SOAP does not have any theoretical basis. However, SOAP uses the same basis as Shampoo, for which the theoretical basis in terms of optimal Kronecker approximation has already been established in previous works [1,2].

    [1] - Gupta et al. 2018, Shampoo: Preconditioned Stochastic Tensor Optimization 2018

    [2] - Morwani et al. 2024, A New Perspective on Shampoo's Preconditioner,

Overall, without proper comparison to other existing second order methods, it is unclear if there is any significant contribution of this work.

**Questions:**

Is the proposed method exactly the same as SOAP without any accumulation? If yes, SOAP is very important as a baseline for verifying if accumulation hurts or helps the performance.

---

> ### Author Response · Authors · 2024-11-25
>
> We thank the reviewer for the feedback and comments.
>
> For the empirical evaluation, we will improve the manuscript based on the reviewer's suggestions.

---

### Official Review · Reviewer_vYVE · 2024-11-03

**Soundness:** 3
**Presentation:** 3
**Contribution:** 2
**Rating:** 5
**Confidence:** 4

**Summary:**

The paper proposes an approach to improving adaptive optimization algorithms for neural networks. Instead of explicitly constructing a non-diagonal preconditioner matrix, the authors propose to transform the gradient to a basis in which the preconditioner is approximately diagonal. The transformation matrix is determined by a periodically computed SVD of the gradient matrix. Following recent work, the authors show that a continuous form of Adam with this algorithmic modification converges to local optima. To demonstrate the superior performance of their method, the authors show results for training a ResNet and vision transformers on ImageNet and pretraining a language model.

**Strengths:**

- Previous work on this class of methods usually motivates the construction of the transformation matrix by computing the eigenbasis of a KFAC/Shampoo preconditioner. This work argues directly with the SVD of the gradient matrix which is equivalent to the eigenvectors of Shampoo’s preconditioner without any accumulation. I think it is a good contribution to explicitly spell out this motivation for the transformation matrix and how it leads to an approximately diagonal preconditioner in the new basis.
- The experimental results are promising, despite their weaknesses (see the next section). If the trend of the results holds consistently, the impact of these methods could be significant because adaptive optimization algorithms are ubiquitous in deep learning.
- The presentation and writing is mostly clear and easy to follow.

**Weaknesses:**

I will describe the weaknesses of each of the three main contributions listed in the introduction (lines 068-078):

1. Regarding the proposed method itself, the paper ignores highly relevant previous work: Liu et al. (2018) [1] and George et al. (2018) [2] also propose to rotate the parameter space in a way that the preconditioner is approximately diagonal. Liu et al. propose this to improve the continual learning method EWC and George et al. aim to improve the optimization performance of K-FAC; they call the resulting method EK-FAC. One difference to this paper is how the transformation matrix is chosen since they use the eigenbasis of the K-FAC preconditioner, which is very closely related to the here proposed transformation matrix (the left- and right-singular vector of the gradient matrix, or equivalently, the eigenvectors of Shampoo’s preconditioner without any accumulation), but not exactly the same. The choice of the transformation matrix proposed in this paper has also been mentioned in Appendix B of Anil et al. (2020) [3] (the only difference being whether we use accumulation to compute Shampoo’s preconditioner), who describe how to adopt the EK-FAC method to use the eigenvectors of Shampoo’s instead of K-FAC’s preconditioner as the transformation matrix.
Finally, the authors mention SOAP by Vyas et al. (2024), which is concurrent work. However, I think the statement that “their approach seems to lack an interpretable framework to explain its effectiveness” (c.f. lines 371-372) is incorrect: since Vyas et al. discuss the related work mentioned here, which already provided a framework to explain the effectiveness of this class of methods, they inherit all the explanations that are valid for any of these works.
I will increase my score if an accurate discussion of all of this previous work is added.

2. Regarding the convergence analysis in section 4, it seems like the stated result is already contained in the cited work by Liang et al. (2024): the case of AdaDiag corresponds to the results in section 4.1+4.2 in Liang et al. (here the transformed gradient is $P_t^T \nabla \mathcal{L}(W_t)$) and the case of AdaDiag++ is described in section 4.3 where the results are generalized to general linear operators (including the explicitly mentioned case where the transformed gradient is $P_t^T \nabla \mathcal{L}(W_t) Q_t$). The choice of the projection matrices differs between the two works since Liang et al. are motivated by memory-efficient optimization, but unless I’m missing something this doesn’t seem to affect the convergence analysis significantly.

3. While the experimental results are promising, they are limited in multiple ways: a) All results seem to focus on either final validation performance or convergence in iterations. However, in the chosen optimization settings it is crucial to evaluate the optimization performance in wall-clock time as well. You mention that the computational overhead is less than 10% in the text, but don't show any experiments for this. b) To substantiate claims that AdaDiag variations “can substantially enhance the convergence speed of modern adaptive optimizers” (abstract), I would expect to see a wider range of benchmarks, beyond the two datasets and three architecture considered here. c) Alternatively, if the claims were restricted to the image classification and language model settings considered here, I would like to see a slightly more thorough evaluation, e.g. runs with multiple random seeds, plots that compare the runs in wall-clock time as well, either more tuning of all methods or a more elaborate justification why you would expect the Adam baseline to be strongly tuned already, and additional ablations, e.g. how the performance depends on the hyperparameter $T$ which is introduced on top of Adam's hyperparameters.

[1] Liu et al. (2018). [Rotate your Networks: Better Weight Consolidation and Less Catastrophic Forgetting](https://arxiv.org/abs/1802.02950)

[2] George et al. (2018). [Fast Approximate Natural Gradient Descent in a Kronecker-factored Eigenbasis](https://arxiv.org/abs/1806.03884)

[3] Anil et al. (2020). [Scalable Second Order Optimization for Deep Learning](https://arxiv.org/abs/2002.09018)

**Questions:**

- Please add an accurate discussion of the related work mentioned in this review.
- Could you please explicitly highlight the difference between your convergence analysis and Liang et al. (2024)?
- You could consider to expand on your experimental results by following some of my suggestions in the previous section.

**Minor comments**

- There is a $\nabla$ symbol missing inside of the EMA in the first two equations in section 2.2.
- In Figure 4 you cut the x-axis of the plots for the two vision transformer experiments at 80 epochs despite training for 300 epochs in total. This first seems to contradict the description of the experiment until you clarify it in the text. Maybe visually show that the run is only partially shown and add this information to the figure's caption, or just show the full plot from the appendix; you could also consider cutting the y-axis at a larger value if you want to improve the visibility of the difference between the methods.

---

> ### Author Response · Authors · 2024-11-25
>
> We thank the reviewer for the very constructive feedback and thoughtful comments.
>
> We appreciate the reviewer's acknowledgment of our contributions. We would like to address the concerns of the reviewer as follows:
>
> 1. For previous works, such as Liu et al. (2018) [1] and George et al. (2018), we thank the reviewer for pointing them out. We agree that these works are relevant, as they also propose diagonal approximations for the preconditioner matrix via projected or rotated transformations. However, while these works all target the Fisher Information Matrix, which is typically implemented in the natural gradient or Laplace approximation family, our work is the first to target the adaptive moment-based preconditioner. We will provide a detailed discussion of these related works in the revised version.
>
> 2.  Regarding the convergence analysis, we inherited the Hamiltonian Descent framework introduced in the cited work by Liang et al. (2024). In their work, to provide the convergence guarantee for arbitrary update rules of the projection matrix $P$, the authors need a mild condition to avoid the degenerate case of $P_t G_t = 0$ while $G_t = 0$ in the invariant set of the system. In our analysis, this assumption is not required when $P$ is a full-rank orthogonal matrix.
>
> 3. For the experimental aspect, we will enhance the manuscript based on the reviewer's suggestions.
>
> For the minor comments, we will edit them in the revised version.

---

> > ### Comment · Reviewer_vYVE · 2024-11-25
> >
> > Thanks for addressing my comments.
> >
> > 1. While it is true that Liu et al. (2018) [1] and George et al. (2018) [2] target the (empirical) Fisher, Anil et al. (2020) [3] also target the adaptive moment-based preconditioner. Their approach should be exactly equivalent to yours if they removed the accumulation and just used the current mini-batch gradient matrix. Also, please remember to address my comment regarding SOAP by Vyas et al. (2024). I encourage you to already update the paper with a discussion of all the related work during the discussion period if you want me to increase my score.
> > 2. Thanks for the clarification. In this case, I view your statement in the paper as a direct corollary of Liang et al. (2024).

---

> > > ### Author Response · Authors · 2024-11-27
> > >
> > > We thank the reviewer for the constructive feedback.
> > >
> > > We have updated the discussion of related works in Section 3.3 of the revised version. We once again thank the reviewer for acknowledging the contributions of our work and remain attentive to the aspects of the paper that can be further improved.

---

> > > > ### Comment · Reviewer_vYVE · 2024-11-30
> > > >
> > > > Thanks for updating Section 3.3. As promised, I have increased my score from 3 to 5. I encourage the authors to work on the empirical evaluation of their method for future revisions of the paper.

---

### Official Review · Reviewer_Nom1 · 2024-11-08

**Soundness:** 3
**Presentation:** 3
**Contribution:** 3
**Rating:** 5
**Confidence:** 4

**Summary:**

The paper proposed a new optimizer called AdaDiag. The key idea is to rotate the preconditioner matrix to a near-diagonal matrix.

**Strengths:**

The topic is important and interesting. The writing is mostly clear. The proposed method makes a lot of sense to me. It is good to see that the idea of rotation can bring good performance gain.

**Weaknesses:**

The proposed method requires extra memory consumption and extra computational overhead brought by SVD. More rigorous discussion on these aspects is needed. See below.

**Questions:**

**My questions and suggestions to improve the script.**

1. **Need more simple examples to motivate.** I suggest the authors do a simple experiments on how AdaDiag works on quadratic functions (with dense Hessian).  This type of experiment will make AdaDiag more convincing.

2.  **Extra computational overhead by SVD.** "While the total computational overhead induced by periodic SVD is negligible (less than 10%), " I kindly disagree this is negligible. Please add more discussions on the wall-clock running time and compare with AdamW rigorously. Right now, it is too hand-waving and informal.

3. **Extra memory.** Please write down the exact memory requirement of AdaDiag and compare to AdamW (e.g., 50% extra memory over Adam's optimizer state?).  Please write down the exact memory footprint (in GB)  on a concrete LLM architecture like Llama. Currently, Table 3 is too abstract and not so informative for the general audience.

4. **Distributed implementation.** Please discuss the challenges of implementing SVD when the weight matrix is shared in >=2 GPUs. Please discuss potential solutions.

5. **Ablation studies.** Please provide more ablation studies on the SVD frequency T. It would be interesting to see how the choice of T affects the performance.

4. What is the difference between AdaDiag and full-rank-Galore?  Need more clarification.

To sum up, I believe the idea of  "rotating the preconditioner" is reasonable and promising. The idea is worth sharing with the community.   However, by judging the paper as a whole,  I think the current script is still rather preliminary.  For instance, the discussions on the weaknesses (e.g., memory and throughput) are too hand-waving and informal. I think it is okay to leave the memory & throughput issue for future investigation,  but as a scientific paper, more serious discussion is needed to justify the current weaknesses.

Overall, I think this is a good paper with a very promising idea, but the current version needs more polishment to become a rigorous scientific paper.

**Some related works:** I suggest the authors add discussions on the following related works.

1. " first-order methods typically require meticulous tuning of hyperparameters to ensure the optimization process can converge to the desired local optima." The work [1] provides extensive evidence that hyperparameters (of Adam) need to be tuned carefully, otherwise the algorithm would diverge. Perhaps the authors can cite this work to ground the claim that " first-order methods typically require meticulous tuning ".

2. "these algorithms have demonstrated more robust empirical results than SGD in various domains and often exhibit better convergence behaviors in practice"  The work [2,3] provides extensive evidence that Adam outperforms SGD on LLMs and investigates why it happens.


3. Section 2 in the work [4] also provides numerical evidence that the Adam preconditioner is more effective when the Hessian is rotated to a near-diagonal matrix.  This also supports the effectiveness of the proposed AdaDiag (++).


[1] Zhang, Y., Chen, C., Shi, N., Sun, R., & Luo, Z. Q. (2022). Adam can converge without any modification on update rules.

[2] Kunstner, F., Yadav, R., Milligan, A., Schmidt, M., & Bietti, A. (2024). Heavy-tailed class imbalance and why adam outperforms gradient descent on language models.

[3] Zhang, Y., Chen, C., Ding, T., Li, Z., Sun, R., & Luo, Z. Q. (2024). Why transformers need adam: A hessian perspective.

[4] Zhang, Y., Chen, C., Li, Z., Ding, T., ... & Sun, R. (2024). Adam-mini: Use fewer learning rates to gain more.

---

> ### Author Response · Authors · 2024-11-25
>
> We thank the reviewer for the constructive feedback and thoughtful comments.
>
> We appreciate the reviewer's recognition of our contributions. We will add the suggestions from reviewers regarding empirical investigations and discussion of related works to the revised version.
>
> Regarding the connection and difference with Galore, we have provided discussions in Section 3.1. Since GaLore emphasizes the low-rank nature of gradient information to improve memory efficiency, using full-rank projection does not logically fit within its framework. This means that identifying the superior performance of full-rank projection and then offering an algorithmic interpretation are key contributions to our work. In particular, the _two-sided_ full-rank projection (AdaDiag++) appears naturally under our framework, which may not be as apparent when viewed through the lens of _full-rank_ GaLore.

---

> > ### Comment · Reviewer_Nom1 · 2024-12-03
> > **Thanks for the response**
> >
> > Thanks for the response and I apologize for the late response. I still believe this paper proposed a very promising and valuable idea for improving optimizers. I sincerely hope the authors can improve the rigor of the paper presentation, and I am really looking forward to the future version.

---

### Meta-Review · Area_Chair_bnD3 · 2024-12-19

**Metareview:**

The paper introduces a novel approach to improving adaptive optimization algorithms for neural networks by transforming the gradient to a space in which the preconditioner becomes approximately diagonal. This transformation is achieved through a periodically computed SVD of the gradient matrix, effectively rotating the preconditioner to a near-diagonal matrix.

While the proposed idea is both innovative and conceptually sound, the paper falls short of demonstrating the practical value of the method. Key limitations include the absence of second-order baselines for comparison, insufficient details regarding distributed implementation, and the significant additional memory and computational overhead introduced by the approach. Furthermore, the paper lacks critical ablation studies to evaluate the contribution of individual components.

Given these shortcomings, along with the uniformly negative feedback from the reviewers, I am inclined to recommend rejecting the paper.

**Additional Comments On Reviewer Discussion:**

During the rebuttal period, all reviewers maintained a negative stance on the paper. The authors did not effectively address the reviewers’ concerns in their rebuttal. No substantial clarifications or new evidence were provided to mitigate the issues raised.

---

### Decision · Program_Chairs · 2025-01-22

Reject